# Restriction of essential amino acids dictates the systemic metabolic response to dietary protein dilution

Yann W. Yap[1,15], Patricia M. Rusu [1,15], Andrea Y. Chan[1], Barbara C. Fam[2], Andreas Jungmann[3,4], Samantha M. Solon-Biet [5], Christopher K. Barlow[6], Darren J. Creek[6,7], Cheng Huang [6], Ralf B. Schittenhelm [6], Bruce Morgan [8], Dieter Schmoll[9], Bente Kiens [10], Matthew D. W. Piper[11], Mathias Heikenwälder[12], Stephen J. Simpson [5], Stefan Bröer [13], Sofianos Andrikopoulos[2], Oliver J. Müller [4,14] & Adam J. Rose [1✉]

Dietary protein dilution (DPD) promotes metabolic-remodelling and -health but the precise nutritional components driving this response remain elusive. Here, by mimicking amino acid (AA) supply from a casein-based diet, we demonstrate that restriction of dietary essential AA (EAA), but not non-EAA, drives the systemic metabolic response to total AA deprivation; independent from dietary carbohydrate supply. Furthermore, systemic deprivation of threonine and tryptophan, independent of total AA supply, are both adequate and necessary to confer the systemic metabolic response to both diet, and genetic AA-transport loss, driven AA restriction. Dietary threonine restriction (DTR) retards the development of obesity-associated metabolic dysfunction. Liver-derived fibroblast growth factor 21 is required for the metabolic remodelling with DTR. Strikingly, hepatocyte-selective establishment of threonine biosynthetic capacity reverses the systemic metabolic response to DTR. Taken together, our studies of mice demonstrate that the restriction of EAA are sufficient and necessary to confer the systemic metabolic effects of DPD.

[1] Department of Biochemistry and Molecular Biology, Metabolism, Diabetes and Obesity Program, Biomedicine Discovery Institute, Monash University, Clayton, VIC 3800, Australia. [2] Department of Medicine (Austin Health), University of Melbourne, Heidelberg, VIC 3084, Australia. [3] Department of Internal Medicine III, University Hospital Heidelberg, Heidelberg, Germany. [4] German Center for Cardiovascular Research (DZHK), Partner sites Kiel and Heidelberg, Germany. [5] Charles Perkins Centre, School of Life and Environmental Sciences, University of Sydney, Sydney, NSW, Australia. [6] Biomedical Proteomics and Metabolomics Facility and the Department of Biochemistry and Molecular Biology, Biomedicine Discovery Institute, Monash University, Clayton, VIC 3800, Australia. [7] Monash Institute of Pharmaceutical Sciences, Monash University, Melbourne, VIC 3052, Australia. [8] Institute for Biochemistry, Centre for Human and Molecular Biology (ZHMB), Saarland University, 66123 Saarbrücken, Germany. [9] Sanofi-Aventis Deutschland GmbH, Industriepark Hoechst, Frankfurt am Main 65926, Germany. [10] Section of Molecular Physiology, Department of Nutrition, Exercise and Sports, Faculty of Science, University of Copenhagen, 2100 Copenhagen, Denmark. [11] School of Biological Sciences, School of Life and Environmental Sciences, Monash University, Clayton, VIC 3800, Australia. [12] Division of Chronic Inflammation and Cancer, German Cancer Research Center, 69120 Heidelberg, Germany. [13] Research School of Biology, Australian National University, Canberra, ACT 0200, Australia. [14] Department of Internal Medicine III, University of Kiel, Kiel, Germany. [16]These authors contributed equally: Yann W. Yap, Patricia M. Rusu. ✉email: adam.rose@monash.edu

The current classification of essential amino acids (EAA) is based on the nutritional requirements for growth and vitality under nil dietary supply of an amino acid (AA)[1]. However, humans rarely face dramatic protein/AA insufficiency, and for the first time in human history, nutritional excesses mean the amount of overweight people outnumber the amount of underweight people on a global scale[2]. This calls for a reconsideration of AA functions in nutrition, now based upon health-related criteria. One approach is dietary protein dilution (DPD), where protein is reduced and replaced by other nutrient sources, and is distinct from caloric restriction[3–6]. Unlike severe protein/AA restriction, which is not compatible with vitality, moderate DPD promotes longevity in multiple species including flies[7–9], rodents[4,10,11], and perhaps humans[12]. Furthermore, DPD also affects health-span and preclinical studies have demonstrated that DPD can retard age-related diseases such as cancer[12,13], type 2 diabetes[14,15], and dyslipidemia/fatty liver disease[16,17]. Notably, dietary protein intake rates are positively related to type 2 diabetes risk as well as all-cause mortality in humans[18,19].

While metabolic health and longevity pursuant to DPD is likely to involve numerous mechanisms, it was recently demonstrated that DPD promotes metabolic and physiological adaptations via constituent AA[14,20,21], which is largely mimicked by a genetic deficiency in intestinal and kidney neutral AA transport[22]. In particular, dietary protein or AA restriction promotes such metabolic remodelling and health by the liver-derived hormone fibroblast growth factor 21 (FGF21)[23], largely by increasing energy expenditure relative to food energy intake[14,17,24–32]. Importantly, DPD also increases FGF21 in humans[14,21,24,33], and this increase is associated with heightened energy expenditure[33] and improved indices of metabolic health[14]. However, whether the simultaneous increase in other non-protein/AA nutrients such as certain carbohydrates[34–36] are required for DPD effects, as well as which particular AA drive this process is currently debated[23,37]. On one hand, restriction of EAA such as the branched chain amino acids[21,38–41] and sulfur-containing AAs[42–45], have been shown to be sufficient in conferring the systemic effects of DPD. On the other hand, others have demonstrated that altered somatic non-EAA metabolism is sufficient and necessary for DPD effects[14,46–49]. Here we attempted to resolve this issue by formally testing which particular amino acids are required to induce the systemic response to DPD and show that specific dietary EAA restriction is both sufficient and necessary to drive the systemic metabolic response to DPD.

## Results

**EAA restriction dictates the metabolic response to DPD.** We previously demonstrated that DPD improves glucose and lipid homoeostasis in obesity and that dilution of amino acids (AA) was sufficient to mimic these effects[14,16,49]. However, it remained to be resolved whether the dilution of protein by substituting with other dietary nutrients (i.e., carbohydrate and fat), and/or which particular dietary AA, were responsible. To examine this, we fed mice a protein-restricted diet (5% energy from casein protein) and compared the effects to a diet where the reduced protein component was replaced with proteinogenic AA at ratios found within casein (see Supplementary Tables 1 and 2 for diet compositions). This experiment demonstrated that AA add-back reversed the effects of protein restriction to reduce serum urea (Fig. 1a), a biomarker of general protein/AA supply, establishing that the addition of purified amino acids functioned physiologically as protein equivalents. Furthermore, AA add-back to a protein-restricted diet reversed the depressed feed efficiency (Fig. 1b), as calculated from higher body mass gain despite lower food intake (Supplementary Fig. 1B, C). This altered feed

efficiency was reflected in an opposite pattern in energy expenditure (EE; Fig. 1c), as gauged from $O_2$ consumption (Fig. 1d) and $CO_2$ production (Fig. 1e) rates, with no differences in respiratory exchange ratio (RER; Supplementary Fig. 1D). The increase in EE with DPD occurred independently from differences in body mass during measurement as judged by ANCOVA (Fig. 1f; adjusted means: NP $0.786 \pm 0.005$ W, LP: $0.955 \pm 0.05$ W, LP + AA: $0.798 \pm 0.04$ W; LP versus NP or LP + AA: $P < 0.001$). The higher EE could not be explained by altered physical activity (Fig. 1g). Consistent with the notion that increased circulating FGF21 is obligatory for the effects of DPD to increase EE[14,16,24,26], we found highly elevated levels of blood plasma FGF21, which were reversed by AA-add back (Fig. 1h). As increased blood FGF21 levels also confer the effects of DPD on improved glucose metabolism[14,49], we also assessed this by measuring an index of fasting insulin sensitivity (ISI(f)) (Fig. 1i) from both fasting glucose (Supplementary Fig. 1E) and insulin (Supplementary Fig. 1F), which correlates well with improved glucose metabolism with DPD[14]. The increased ISI(f) with DPD was indeed completely reversed with AA add-back (Fig. 1i).

While this experiment demonstrated that AA could be a necessary component of these effects, dietary carbohydrate was concomitantly manipulated, and could potentially explain the responses as FGF21 is affected by certain dietary carbohydrates[31,34–36]. In addition, the specific AA conferring the effects of DPD were not identified. As the liver is an essential organ involved in "sensing" DPD[14,26], we initially investigated the hepatic portal vein (Fig. 2a and Supplementary File 1) and liver (Fig. 2b and Supplementary File 1) AA levels in response to DPD. While most nutritional[1] EAA were lower in the portal vein plasma and liver with DPD, certain non-essential AAs (NEAA) such as Asn, Pro, Glu, and Tyr were also affected meaning that we had to take a more broader approach than just focussing on one class of AA. In order to examine this, we conducted a study where we manipulated the EAA and NEAA specifically, on the basis of nutritional definitions[1,50], and in some diets topping up the alternate source of AA to keep the total AA supply constant without altering dietary fat or carbohydrate supply (Fig. 2c). Of note, NEAA supply positively correlated with urinary and blood serum urea more so than EAA supply (Fig. 2d and Supplementary Fig. 2A), and there were non-binary relationships between EAA/NEAA supply and feed efficiency (Fig. 2e and Supplementary Fig. 2B, C). However, EAA restriction fully conferred the effects of DPD on EE (Fig. 2f) and serum FGF21 levels (Fig. 2g). In particular, serum FGF21 levels were inversely related to total EAA supply (Fig. 2g).

In this study, we also conducted glucose tolerance tests to further examine diet effects on whole-body metabolism. The blood glucose excursion (Fig. 2h) and related area under the curve (Supplementary Fig. 2D) was directly related to EAA supply, with lower glucose levels found with EAA restriction. Similar results were seen for that of insulin (Fig. 2i and Supplementary Fig. 2E). From fasting glucose (Supplementary Fig. 2F) and insulin (Supplementary Fig. 2G) values, we could calculate various indices of glucose metabolic control such as the ISI(f) (Fig. 2j) and HOMA-IR (Supplementary Fig. 2H). In addition, we calculated the product of the glucose and insulin AUCs (Supplementary Fig. 2I). Importantly, there were very close correlations between the indices determined from fasting glucose and insulin compared with those determined from the glucose tolerance tests (Supplementary Fig. 2J–L). All of these indices highlighted that glucose metabolism/insulin sensitivity is heightened with dietary EAA restriction.

**Thr and Trp restriction is necessary for DPD effects.** By way of follow-up, we then sought to determine which particular EAA

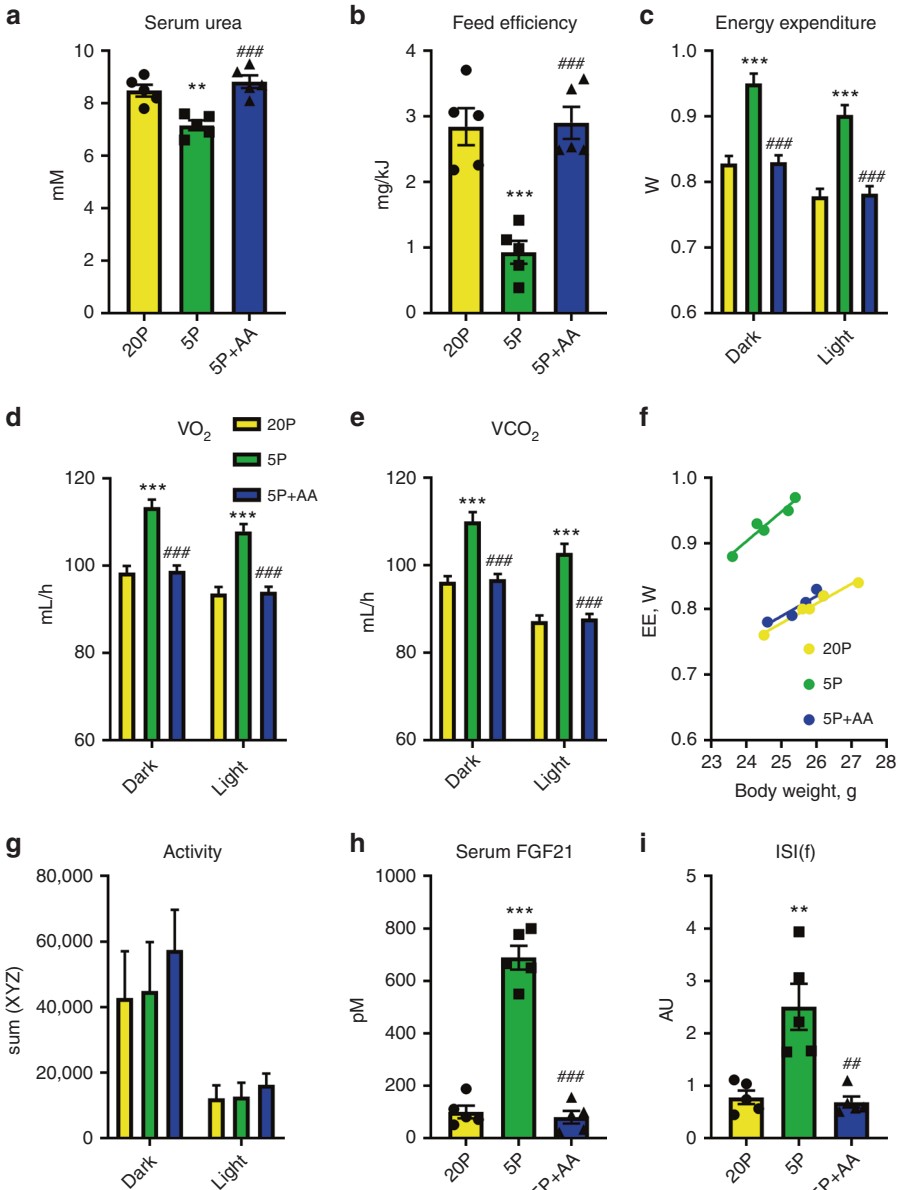

**Fig. 1 Dietary amino acids are required for the systemic metabolic effects of dietary protein dilution. a** Serum urea levels of mice in response to a 3-week treatment with diets containing 20% energy from protein (20P), 5% energy from protein (5P), and 5% energy from protein and 15% energy from amino acids to match that of 20P. Data are mean and SEM; $n = 5$ individual mice per group. Data were analysed by one-way ANOVA with Holm–Sidak post-hoc tests. Different than 20P: *$P < 0.05$, **$P < 0.01$, ***$P < 0.001$. Different than 5P: #$P < 0.05$, ##$P < 0.01$, ###$P < 0.001$. **b** Feed efficiency of mice as in (**a**). **c** Energy expenditure over the different day phases of mice as in (**a**). **d** The rate of $O_2$ consumption ($VO_2$) over the different day phases of mice as in (**a**). **e** The rate of $CO_2$ production ($VCO_2$) over the different day phases of mice as in (**a**). **f** Scatter plot of energy expenditure (EE) versus body weight of mice as in (**a**). **g** Physical activity as assessed by laser beam breaks across three physical dimensions (sumXYZ) over the different day phases of mice as in (**a**). **h** Serum fibroblast growth factor 21 (FGF21) levels of mice as in (**a**). **i** Insulin sensitivity index during fasting (ISI(f)) of mice as in (**a**).

could confer these effects and selected subgroups based upon known biochemical features[50]. In particular, we chose one subgroup based on their classification as ones which cannot be synthesised by any possible precursor within the mammalian metabolic network (i.e., strictly metabolically essential; Lys, Thr, Trp), the branched-chain AA (i.e., Ile, Leu, Val), and the remaining three (i.e., His, Met, Phe). We then conducted studies where we added these EAAs back to the low EAA diet and notably it was only the strictly metabolically essential AA, namely Lys, Thr, and Trp, which were necessary to confer the systemic metabolic effects to total EAA deprivation (Fig. 3a–d and Supplementary Fig. 3A–E).

In an attempt to investigate whether a single one of these EAAs could confer the effects of dietary EAA restriction (DEAR), we then individually added back either Lys, Thr, or Trp to the EAA restricted diet and could demonstrate that no single one of these was necessary for the effects of DEAR (Supplementary Fig. 3F–I). By logical deduction, this meant that restriction of at least two, and perhaps all three, of these EAA were necessary for the full effects of DEAR. To initially test this, we then conducted a study where we singly restricted Lys, Thr, or Trp, at levels matching those found in the complete AA restriction, and could demonstrate that deprivation of either Thr or Trp, but not Lys, was sufficient to mimic the effects of DEAR (Fig. 3e–h and

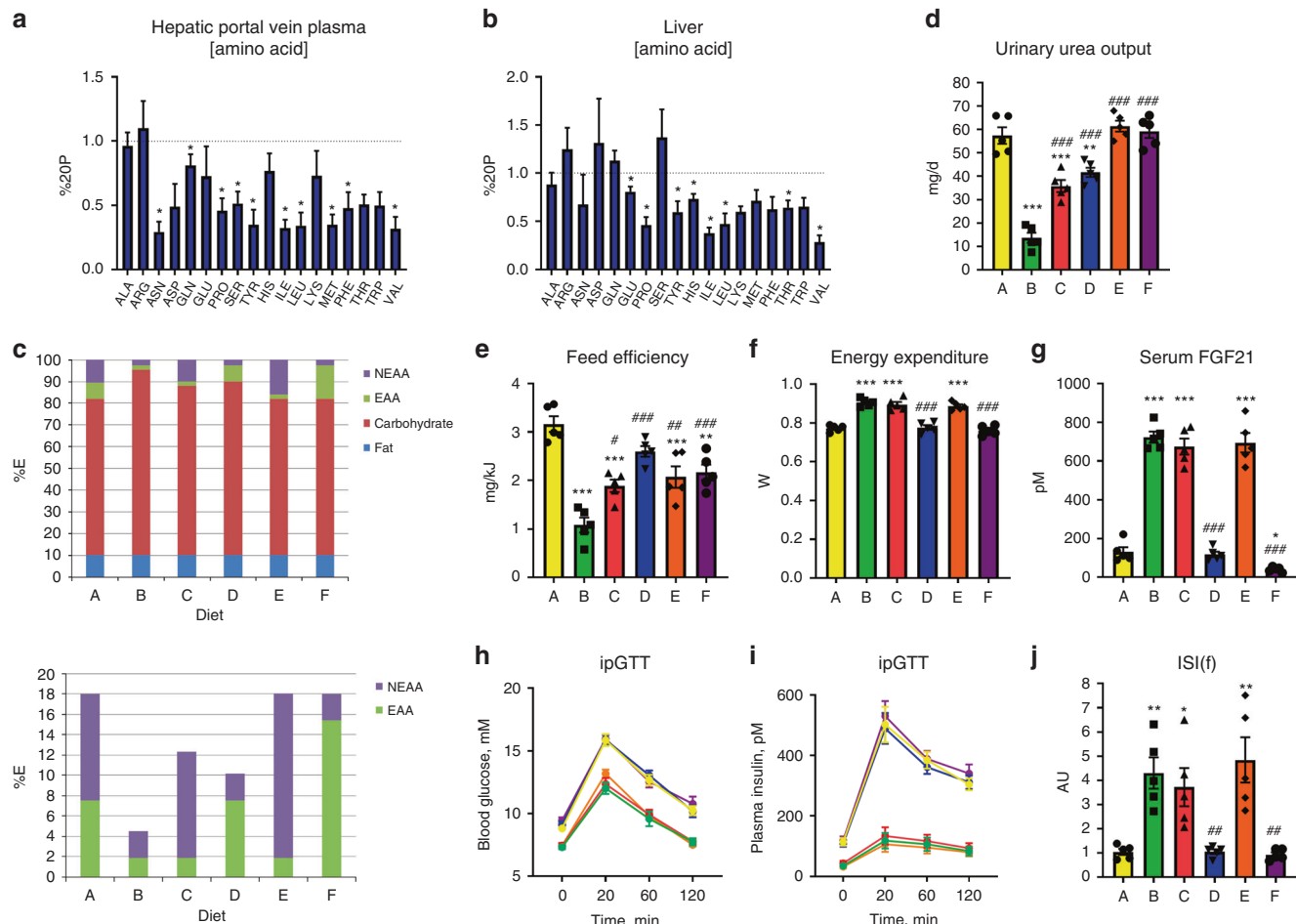

**Fig. 2 Dietary essential amino acid restriction, independent from non-essential amino acid or carbohydrate supply, dictates the systemic metabolic response to dietary protein dilution. a** Hepatic portal vein amino acid levels in response to refeeding a low-protein diet following a week of diet adaptation. Data (mean and SEM) are represented as a proportion of the control group fed a normal protein diet (20%P). $n = 5$/group derived from samples of individual mice. Data were analysed by Student's $t$-test. Different than 20%P, *$P < 0.05$. **b** Liver tissue amino acid levels in response to refeeding a low-protein diet following a week of diet adaptation. Data are represented as a proportion of the control group fed a normal protein diet (20%P). $n = 5$/group derived from samples of individual mice. Data were analysed by Student's $t$-test. Different than 20%P, *$P < 0.05$. **c** Nutrient source breakdown as % contribution to total energy for the experimental groups. EAA: essential amino acids. NEAA: non-essential amino acids. **d** Urinary urea output of mice in response to a 3-week treatment with diets as per the protocol of SF1A containing nutrient energy sources as in (**c**). Data are mean and SEM; $n = 5$ individual mice per group. Data were analysed by one-way ANOVA with Holm–Sidak post-hoc tests. Different than 20P: *$P < 0.05$, **$P < 0.01$, ***$P < 0.001$. Different than 5P: #$P < 0.05$, ##$P < 0.01$, ###$P < 0.001$. **e** Feed efficiency of mice as in (**d**). **f** Energy expenditure of mice as in (**d**). **g** Serum fibroblast growth factor 21 (FGF21) levels of mice as in (**d**). **h** Blood glucose levels during an intraperitoneal glucose tolerance test (ipGTT) of mice as in (**d**). **i** Plasma insulin levels during an intraperitoneal glucose tolerance test (ipGTT) of mice as in (**d**). **j** Insulin sensitivity index during fasting (ISI(f)) of mice as in (**d**).

Supplementary Fig. 3J–N). Importantly, this finding was independently supported in separate experiments whereby a diet low in Met, Thr, and Trp induced feed inefficiency and increased serum FGF21 similar to DPD (Supplementary Fig. 3O–R).

To test the necessity of the deprivation of these two EAA for the effects of DEAR, we then conducted studies with selective add-back of these two EAA in the background of a low total AA supply, and could demonstrate that deprivation of both Thr and Trp were required for the systemic metabolic effects of DEAR (Fig. 3i–l and Supplementary Fig. 3S–W). In summary, in the background of total dietary protein/AA restriction simultaneously low levels of Thr and Trp are required for the full effects, whereas restriction of either Thr or Trp individually can mimic the majority of effects of DPD.

Our prior studies were done on male mice from 8 weeks of age, which could potentially limit the applicability of our findings, as female mice are known to respond differently to dietary challenges[51–53], and such young mice are still growing and thus

may differ in dietary AA requirements compared with adult mice. Hence, we tested several diets used previously (Figs. 1–3) on 6-month-old male and female mice for a longer time frame (i.e., 8 weeks) to assess potential differences (Fig. 4). Moreover, we tested the effects of total dietary AA restriction (LAA), as well as EAA (LEAA) and threonine (LT) restriction with matched total AA supply, on various parameters such as body composition, metabolic efficiency, and indices of metabolic health. Of note, while LAA and LEAA caused weight loss (LAA) or weight stabilisation (LEAA) (Fig. 4a, b), mostly reflected as lean mass loss, and there was no lean mass loss with LT (Supplementary Fig. 4A, B). This was reflected by end-point tissue weights, with both skeletal muscle and heart weights showing a similar trend to that of lean mass measured by MRI (Fig. 4c, d). Concerning fatness, mice on all diets gained fat mass, but there were no statistically significant differences between diets as assessed by MRI, with the exception of males subjected to LT (Fig. 4a, b). However, when individual adipose tissue depots were assessed,

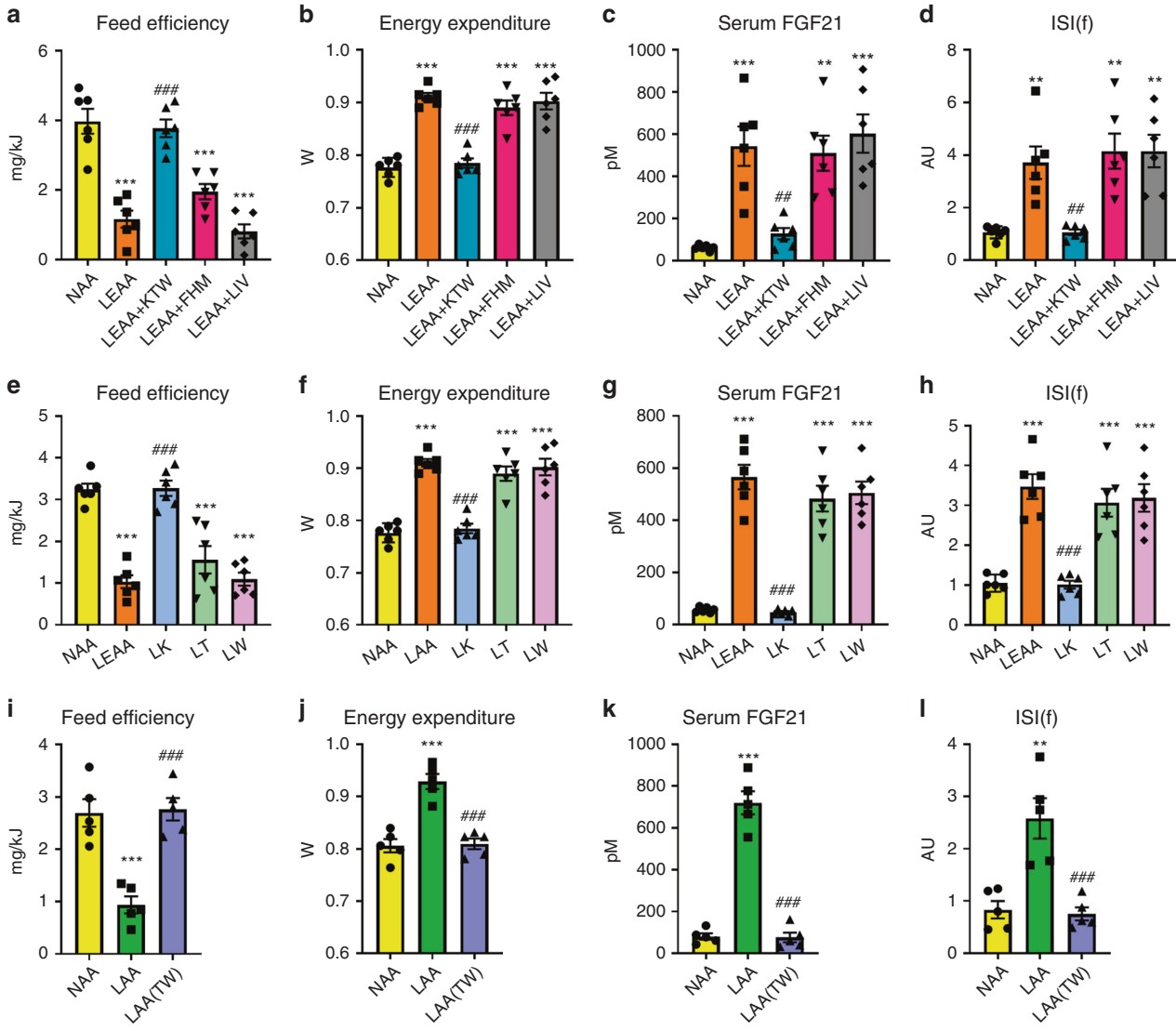

**Fig. 3 Certain essential amino acids including threonine and tryptophan are sufficient and necessary for the systemic metabolic effects of dietary protein dilution. a** Feed efficiency of mice in response to a 3-week treatment with diets containing 18% from amino acids (normal amino acid; NAA), 4.5% essential AA (LEAA; as of diet E in Fig. 2c), and LEAA supplemented with either lysine, threonine, and tryptophan (LEAA + KTW), phenylalanine, histidine, and methionine (LEAA + FHM), or isoleucine, leucine, and valine (LEAA + ILV), all with other AA equally adjusted to give 18% AA in total. Data are mean and SEM ($n = 6$ individual mice per group). Data were analysed by one-way ANOVA with Holm–Sidak post-hoc tests. Different than diet NAA: *$P < 0.05$, **$P < 0.01$, ***$P < 0.001$. Different than diet LEAA: #$P < 0.05$, ##$P < 0.01$, ###$P < 0.001$. **b** Energy expenditure of mice as in (**a**). **c** Serum fibroblast growth factor 21 (FGF21) levels of mice as in (**a**). **d** Insulin sensitivity index during fasting (ISI(f)) of mice as in (**a**). **e** Feed efficiency of mice in response to a 3-week treatment with diets containing 18% from amino acids (AA; NAA), 4.5% essential AA (LEAA; as of diet E in Fig. 2c), and diet singly with restricted amounts of lysine (LK), threonine (LT), and tryptophan (LW), all with other AA equally adjusted to give 18% AA in total. Data are mean and SEM; $n = 6$ individual mice per group. Data were analysed by one-way ANOVA with Holm–Sidak post-hoc tests. Different than diet NAA: *$P < 0.05$, **$P < 0.01$, ***$P < 0.001$. Different than diet LEAA: #$P < 0.05$, ##$P < 0.01$, ###$P < 0.001$. **f** Energy expenditure of mice as in (**e**). **g** Serum FGF21 levels of mice as in (**e**). **h** ISI(f) of mice as in (**e**). **i** Feed efficiency of mice in response to a 3-week treatment with diets containing 18% from amino acids (normal amino acid; NAA), 4.5% AA (LAA; as of diet B in Fig. 2c), and LAA supplemented with threonine and tryptophan while keeping total AA at 4.5% (LAA(TW)). Data are mean and SEM ($n = 5$ individual mice per group). Data were analysed by one-way ANOVA with Holm–Sidak post-hoc tests. Different than NAA: *$P < 0.05$, **$P < 0.01$, ***$P < 0.001$. Different than LAA: #$P < 0.05$, ##$P < 0.01$, ###$P < 0.001$. **j** Energy expenditure of mice as in (**e**). **k** Serum FGF21 levels of mice as in (**e**). **l** ISI(f) of mice as in (**e**).

both perigonadal and subcutaneous fat depots were lower in male, but not female, mice subjected to dietary AA/EAA restriction (Fig. 4c, d). Concerning feed efficiency, the responses were similar to that of the changes in body weight, with dietary AA/EAA restriction promoting consistently reduced feed efficiency regardless of sex (Fig. 4e, f). However, LT feeding produced a consistently higher feed efficiency versus other AA restriction groups (Fig. 4e, f), probably owing to the less

pronounced effects on lean mass, as there was an equal increase in EE (Fig. 4g, h) and food E intake (Supplementary Fig. 4C, D) in all groups subjected to AA/EAA restriction regardless of sex.

Similar to the food E intake and EE responses, serum FGF21 levels were equally higher in mice subjected to AA/EAA restriction, again regardless of sex (Fig. 4i, j). Glucose metabolism also showed a similar response, with an equally higher insulin sensitivity index (Fig. 4k, l), as discerned from fasting blood

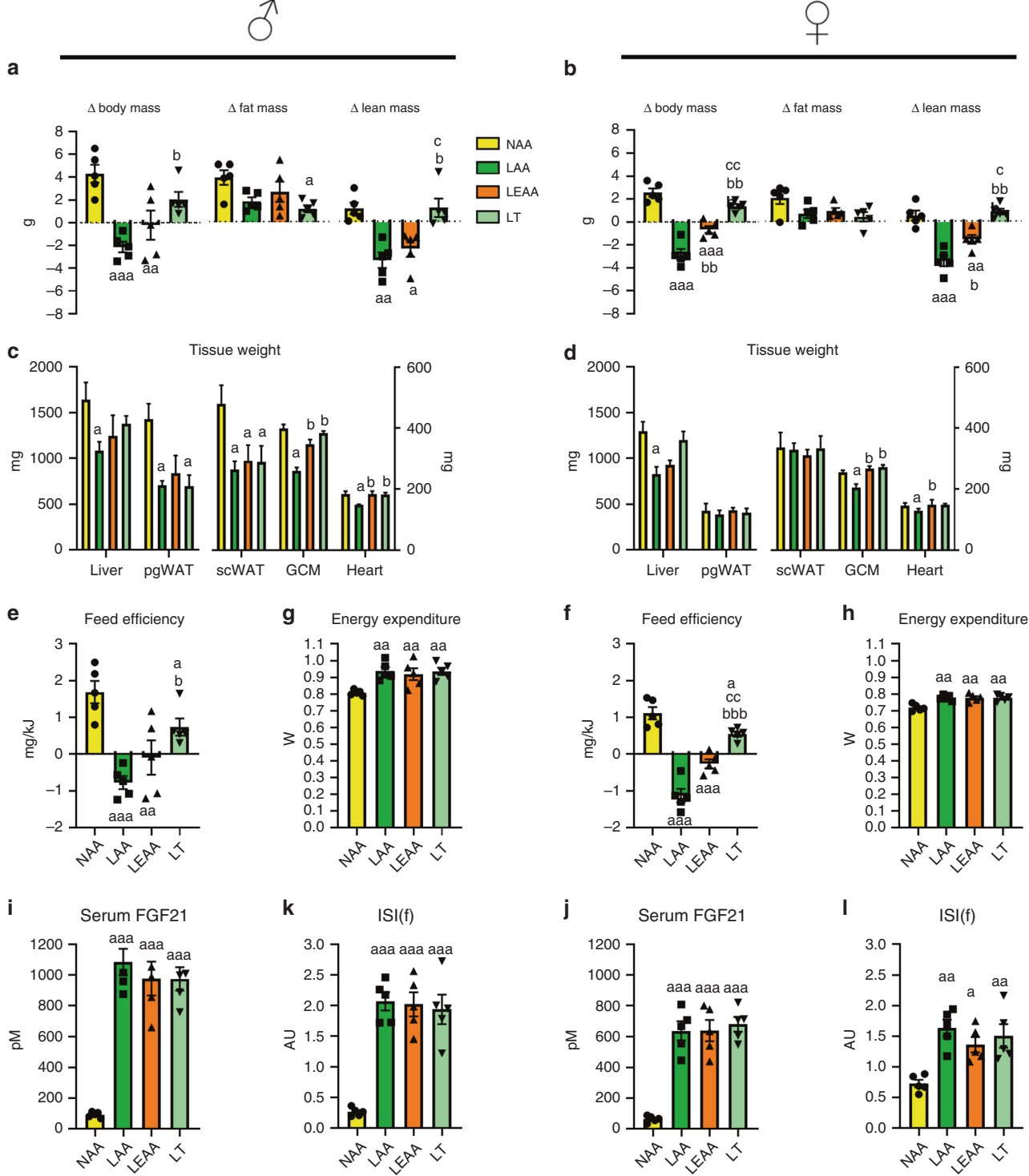

**Fig. 4 The systemic metabolic response to dietary AA restriction is conserved in mature male and female mice. a** The change in body, fat, and lean mass of 6-month-old male (shown left) mice in response to an 8-week treatment with diets containing 18% from amino acids (normal amino acid; NAA), 4.5% AA (LAA; as of diet B in Fig. 2c), 4.5% essential AA (LEAA; as of diet E in Fig. 2c), and a diet low in threonine but with matching total AA to NAA (LT). Data are mean and SEM ($n = 5$ individual mice per group). Data were analysed by one-way ANOVA with Holm–Sidak post-hoc tests. Different than NAA: [a]$P < 0.05$, [aa]$P < 0.01$, [aaa]$P < 0.001$. Different than LAA: [b]$P < 0.05$, [bb]$P < 0.01$, [bbb]$P < 0.001$. Different than LEAA: [c]$P < 0.05$, [cc]$P < 0.01$, [ccc]$P < 0.001$. **b** The change in body, fat, and lean mass of 6-month-old female (shown left) mice treated as in (**a**). **c** Tissue weights of mice at the end of the treatment as in (**a**). **d** Tissue weights of mice at the end of the treatment as in (**b**). **e** Feed efficiency during the 8-week treatment of mice as in (**a**). **f** Feed efficiency during the 8-week treatment of mice as in (**b**). **g** Energy expenditure of mice as in (**a**). **h** Energy expenditure of mice as in (**b**). **i** Serum fibroblast growth factor 21 (FGF21) levels of mice as in (**a**). **j** Serum FGF21 of mice as in (**b**). **k** Insulin sensitivity index during fasting (ISI(f)) of mice as in (**a**). **l** ISI(f) of mice as in (**b**).

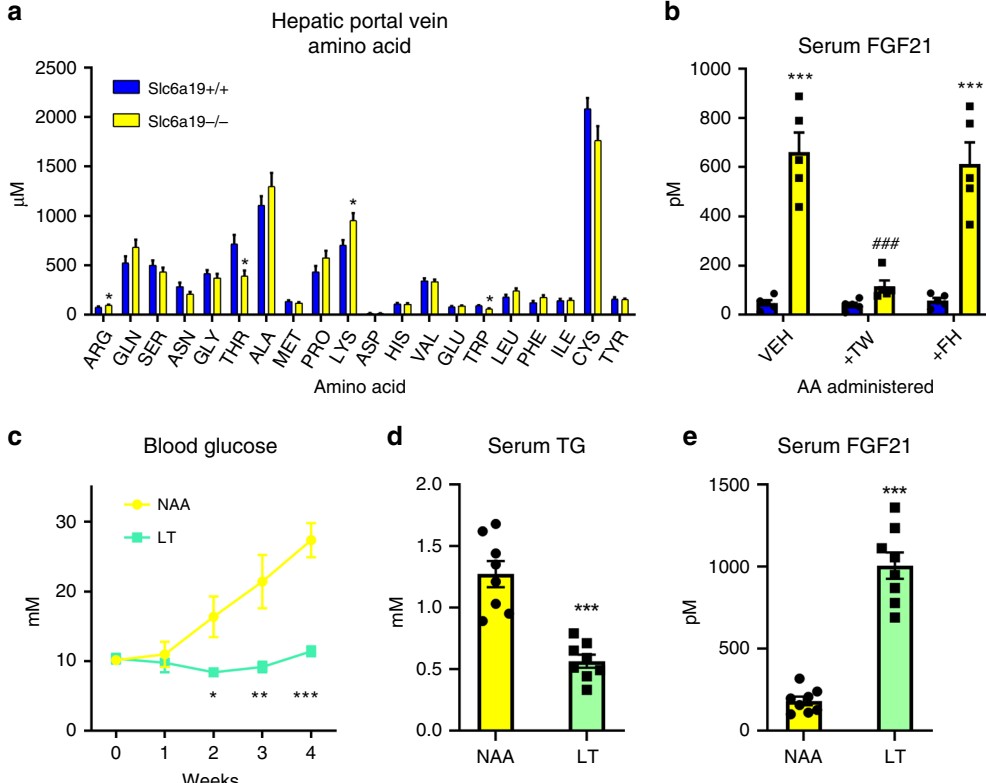

**Fig. 5 Threonine restriction is a common feature of other models of systemic AA restriction and retards obesity-induced metabolic dysfunction in mice. a** Hepatic portal vein serum amino acid (AA) concentrations in Slc6a19 knockout (−/−) or wildtype (+/+) littermate mice in the refed state on a standard control diet. Data are mean and SEM; $n = 5$ individual mice per group group. Data were analysed by a Student's $t$-test. Different than +/+: *$P <$ 0.05. **b** Plasma FGF21 levels from Slc6a19 knockout (−/−) or wildtype (+/+) littermate mice in the refed state following intraperitoneal administration of the amino acids threonine and tryptophan (TW), phenylalanine and histidine (FH), or vehicle (VEH: saline solution) on a standard control diet. Data are mean and SEM; $n = 5$ individual mice per group. Data were analysed by a two-way repeated measures ANOVA. Different than +/+: *$P <$ 0.05, **$P <$ 0.01, ***$P <$ 0.001. Different than VEH: #$P <$ 0.05, ##$P <$ 0.01, ###$P <$ 0.001. **c** Blood glucose levels during a 4-week treatment of New Zealand Obese mice fed diets containing 18% from amino acids (normal amino acid; NAA) or a diet low in threonine but with matching total AA to NAA (LT). Data are mean and SEM; $n = 8$ individual mice per group. Data were analysed by one-way repeated measures ANOVA. Different than NAA: *$P <$ 0.05, **$P <$ 0.01, ***$P <$ 0.001. **d** Serum triglyceride (TG) levels at the end of mice at the end of treatment as in (**c**). Data were analysed by a Student's $t$-test. Different than NAA: *$P <$ 0.05, **$P <$ 0.01, ***$P <$ 0.001. **e** Serum fibroblast growth factor 21 (FGF21) levels of mice at the end of treatment as in (**c**).

glucose (Supplementary Fig. 4E, F) and plasma insulin (Supplementary Fig. 4G, H), in groups subjected to dietary AA/EAA restriction, again regardless of sex.

As DPD also affects lipid metabolism[14,17] and the IGF1 axis[12,14], we also assessed these. Serum triglyceride levels were lower with dietary AA/EAA restriction, and this was more pronounced in male than female mice (Supplementary Fig. 4I, J). Serum IGF1 levels were substantially lower with total AA restriction, but not with dietary Thr restriction, with dietary EAA restriction producing an intermediary response, particularly in male mice (Supplementary Fig. 4K, L).

To reinforce the findings that only certain EAA such as Thr and Trp are important for the effects of dietary EAA restriction, we then wanted to confirm the importance of these EAA in other situations. To this end, we utilised a mouse model with intestinal and renal neutral amino acid transport deficiency (i.e., B⁰AT1/Slc6a19 knockout mouse), which exhibits metabolic features akin to dietary protein/AA restriction[22]. Initially, we characterised the hepatic portal vein AA profile, and could demonstrate that while there were higher levels of certain AA such as Arg and Lys, there was substantially lower levels of Thr and Trp (Fig. 5a). In order to test the involvement of these AA in the phenotype of these mice, we conducted an acute AA add-back study, and could demonstrate that systemic add-back (by

intraperitoneal administration) of Thr and Trp, but not His and Phe, could reverse the upregulation of blood plasma FGF21 levels in these mice (Fig. 5b), highlighting the importance of these two EAA in the response to dietary AA restriction.

**Thr restriction promotes metabolic health via liver FGF21**. As we have previously shown that dietary protein or AA dilution can retard the development of obesity-related metabolic dysfunction, we then tested whether selective Thr restriction (LT), without total AA restriction or altered dietary carbohydrate supply, can mimic these effects in a mouse model of obesity-related metabolic dysfunction, the New Zealand Obese (NZO) mouse. Of note, similar to our prior findings of NZO mice subjected to dietary protein/AA dilution[14,49], there was no effects of LT on body mass development and liver and adipose tissue weights (Supplementary Fig. 5A, B). However, LT completely retarded the development of hyperglycaemia (Fig. 5c), reduced hypertriglyceridemia (Fig. 5d), and increased serum FGF21 levels (Fig. 5e), in this model. Altogether, these results highlight that dietary Thr is a common EAA mediating effects of dietary AA restriction across mouse models, and that dietary Thr restriction can affect positive health outcomes in obesity.

As FGF21 was commonly affected by dietary EAA restriction (Figs. 2–5), and liver-derived FGF21 conveys the metabolic

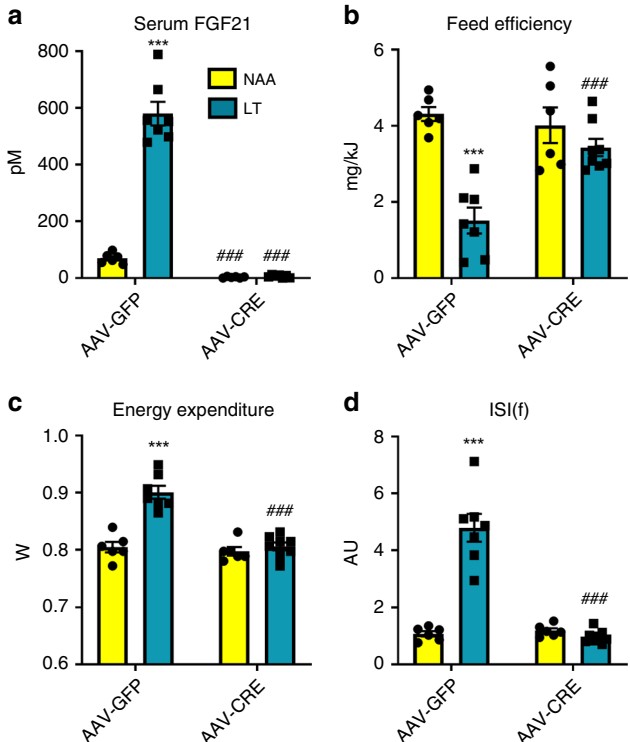

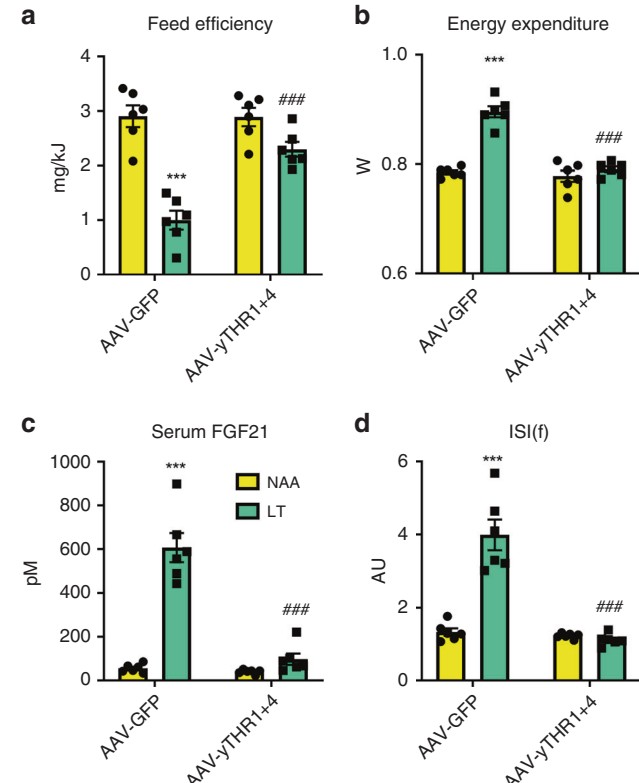

**Fig. 6 Liver-derived fibroblast growth factor 21 is necessary for the systemic metabolic remodelling with dietary threonine restriction.**
**a** Serum fibroblast growth factor 21 (FGF21) levels of Fgf21fl/fl mice at the end of an 8-week treatment with diets containing 18% from amino acids (AA; NAA) or low threonine with other AA equally adjusted to give 18% AA in total (LT); with pre-treatment with adeno-associated viruses to express Cre-recombinase (AAV-CRE) or green fluorescent protein (AAV-GFP) in an hepatocyte-selective manner. Data are mean and SEM ($n = 6$ NAA × AAV-GFP; $n = 7$ LAA AAV-GFP; $n = 6$ LT × AAV-CRE; $n = 8$ LT × AAV-CRE). Data were analysed by two-way ANOVA with Holm–Sidak post-hoc tests. Different than NAA: *$P < 0.05$, **$P < 0.01$, ***$P < 0.001$. Different than AAV-GFP: #$P < 0.05$, ##$P < 0.01$, ###$P < 0.001$. **b** Feed efficiency during the 8-week treatment of mice as in (**a**). **c** Energy expenditure of mice as in (**a**). **d** Insulin sensitivity index during fasting (ISI (f)) of mice as in (**a**).

**Fig. 7 Enforced hepatic threonine biosynthetic capacity reverses the systemic metabolic effects to dietary threonine restriction. a** Feed efficiency of mice in response to a 3-week treatment with diets containing 18% from amino acids (normal amino acid; NAA; yellow bars) and a diet with restricted amounts of threonine (LT; green bars), following prior treatments with adeno-associated viruses to transduce the liver to express yeast threonine biosynthetic enzymes (AAV-yTHR1 + THR4) or a negative control (AAV-GFP). Data are mean and SEM ($N = 6$ individual mice per group). Data were analysed by two-way ANOVA with Holm–Sidak post-hoc tests. Different than diet NAA: *$P < 0.05$, **$P < 0.01$, ***$P < 0.001$. Different than AAV-GFP: #$P < 0.05$, ##$P < 0.01$, ###$P < 0.001$. **b** Energy expenditure of mice as in (**a**). **c** Serum fibroblast growth factor 21 (FGF21) levels of mice as in (**a**). **d** Insulin sensitivity index during fasting (ISI(f)) of mice as in (**a**).

remodelling with DPD[14,24,26] and methionine restriction[42], we tested the requirement of liver-derived FGF21 in the systemic metabolic remodelling with LT. As such, we administered adeno-associated viruses to express Cre-recombinase (AAV-CRE) to Fgf21 floxed mice to silence Fgf21 expression/secretion in a liver/hepatocyte-specific manner. Indeed, serum levels of FGF21 were absent in groups treated with AAV-CRE (Fig. 6a). Hepatocyte Fgf21 silencing retarded the depressed feed efficiency with LT (Fig. 6b), mostly owing to a complete abrogation of the effects on reduced body mass gain with LT (Supplementary Fig. 6A, B), with blunted effects of LT feeding on fat tissue (Supplementary Fig. 6C). In addition, the effects of LT to increase energy expenditure (Fig. 6c and Supplementary Fig. 6D) and food energy intake (Supplementary Fig. 6E) were also abrogated with liver Fgf21 silencing. In congruence, the improved glucose metabolism (Fig. 6d and Supplementary Fig. 6F, G) and reduced serum triglyceride levels (Supplementary Fig. 6H) with LT were also abrogated with liver/hepatocyte Fgf21 loss. In summary, the systemic metabolic remodelling with LT requires liver-derived FGF21.

**Enforced liver Thr biosynthesis reverses Thr restriction effects.** Although EAA systemic metabolic turnover is intimately related

to genome-wide exome EAA abundance[54], the major metabolic fate of EAA may not be for protein synthesis[50]. Thus, we sought to test whether it is the inability to synthesise the strictly EAA that characterises them as the most limiting and thus required for the effects of DEAR. To this end we produced adeno-associated viruses to express the yeast Thr biosynthetic enzymes (i.e., Thr1 and Thr4) in the liver hepatocytes of mice and subjected them to LT. Using high-resolution mass spectrometry, the yeast proteins THR1 and THR4 were confidently identified in mouse liver extracts overexpressing THR1 and THR4, but not of those expressing GFP (Supplementary Table 3 and Supplementary File 2). Liver-specific expression was also confirmed by qPCR. Importantly, the reduction of liver Thr concentration was reverted with liver-specific enforcement of de novo threonine biosynthesis (Supplementary Fig. 7A and Supplementary Table 4). Strikingly, the effects of dietary Thr restriction on metabolic efficiency, serum FGF21, and glucose metabolism were completely reversed by artificially increasing the liver Thr de novo biosynthesis (Fig. 7a–d and Supplementary Fig. 7B–F).

## Discussion

DPD promotes metabolic health by inducing the hepatokine fibroblast growth factor 21 (FGF21)[14,16], but the precise

nutritional components driving this response are not fully defined. In particular, in order to keep total caloric supply neutral, which is an important consideration in diet–phenotype interactions[3,5], another macronutrient source must be concomitantly increased. In former studies[5,14,24], this was typically achieved by increasing dietary carbohydrates, which may be an independent driver of increased FGF21[34,35]. DPD can also increase food intake, but particularly on low protein–high-carbohydrate diets, this increased food intake is surpassed by heightened energy expenditure, promoting a situation of systemic metabolic inefficiency[14,24]. However, on a low-protein–high-fat diet, mice gain more body weight[4,14], which may[4] or may not[14,16] promote worsened metabolic health. This weaker response to differing diluting nutrients in DPD is perhaps due to the absence of relatively weaker satiating effects of lipids versus protein or carbohydrate[55,56]. Nevertheless, pair-feeding studies have shown that the DPD-driven heightened energy expenditure and FGF21 can be uncoupled from food intake[32], indicating that the major driver of metabolic inefficiency is not due to food energy ingestion.

Importantly, our studies here (Figs. 1–7), as well as previous studies of amino acid (AA) restriction[23,37], have conclusively shown that a restriction of dietary protein/AA per se can induce the systemic metabolic response to DPD. This occurs even when total dietary AA enrichment is held constant (Figs. 1–7), and total protein N supply and oxidation is not affected (Fig. 2d). Nevertheless, there are important relationships between total energy and N intake[6,57], and deciphering the roles of certain AAs and FGF21 in this context will be interesting. On another note, even though dietary protein as a nutrient can be digested and absorbed as AA or short peptides, our data reinforce that dietary AA supply is paramount under conditions of dietary protein restriction. Importantly, metabolomic studies of postprandial plasma amino acid levels in Slc6a19 null mice, which mimic the effects of DPD, also revealed a prominent effect on Thr and Trp when fed standard laboratory chow[58]. By contrast, the ablation of intestinal peptide transport does not mimic the effects of DPD[59].

So could restriction of certain AA confer this response? There are several studies showing that dietary EAA restriction is sufficient to induce many of the responses to DPD[20,37]. On the other hand, there are several lines of evidence that restriction of systemic non-EAA (NEAA) supply to the liver influences FGF21 and systemic metabolism[14,46–48]. Thus, with the evidence that both EAA and NEAA supply (i.e., hepatic portal vein) and liver EAA and NEAA levels are affected by DPD (Fig. 2), we sought to systemically investigate the role of EAA and NEAA restriction. Through a series of studies, we empirically determined that EAA, and in particular the EAA Thr and Trp, are necessary and sufficient to induce the systemic metabolic response to DPD (Figs. 2–7), independent of total dietary AA supply. Importantly, this validates a prior study predicting that Thr is the most limiting AA from AA exome matching under a casein-based diet[20] as well as a recent study demonstrating that the hyperphagia due to dietary casein/AA restriction relates to Thr and/or Trp supply[60]. However, here we demonstrate that both Thr and Trp are equally limiting in the casein-based diet through empirical investigation, both by dietary AA restriction and by using a genetic model of AA-transport deficiency (Fig. 5). While this may seem contrary to those studies which have shown that restriction of certain EAA such as the branched chain AA (BCAA) or sulfur-containing AA (SCAA) are sufficient to mimic systemic metabolic response to DPD[37], this may be explained by titration thresholds dictated by ratios of dietary supply and somatic demand. In particular, with regard to titration, basing the deprivation levels on the 5%E casein diet simply identifies those AA most limiting in this natural protein source, and that if we also restricted an AA to a

theoretical level below which they would be limiting according to some somatic constraint[20], then this would likely trigger a similar response. Indeed, even though Lys is an abundant AA in the milk protein casein, it is the most limiting EAA in protein derived from maize (i.e., zein)[61], and the dietary supply of many EAAs are reduced in health-promoting vegan diets[62]. Nonetheless, it is known that certain EAA can be synthesised/spared by metabolism of precursors that can be mobilised within the body and/or supplied by diet[50]. In particular, systemic responses to restriction of the EAA Met can be alleviated by dietary Cys driven Met sparing[63–65]. In addition, restriction of all three BCAA, but not leucine alone, recapitulates many of the features of total AA restriction[21]. This metabolic compensation between BCAA is perhaps explained by that the metabolism of the three BCAA is linked by a single transaminase reaction[66], with rapid in vivo metabolism of BCAA, of which the liver is a major contributor[67], despite the absence of a BCAA transaminase in hepatocytes[66]. Thus, we postulate that if one of the BCAA becomes limiting it can be spared by the other BCAAs, and their metabolism, within the tissue and between tissues. In support of the concept of "metabolic sparing", liver refurbishment of de novo biosynthesis of Thr abated the effects of dietary Thr restriction (Fig. 7). Thus we propose that it is the strictly metabolically EAA, namely Lys, Thr, and Trp, which have the potential to be most limiting in a particular diet. Nonetheless, the differences in the sensitivity of AA restriction requires careful studies of nutritional EAA titration, especially as our studies were fully based on AA supply from a single nutritional source (i.e., the milk protein casein). This is a clear direction for future studies. In addition, further studies are required to carefully dissect which EAA drive (mal)-adaptive processes during DPD, and in particular which do so under circumstances of altered somatic AA metabolism such as age-related disease, infection, or pregnancy. Furthermore, as our studies were conducted in mice the relevance of our findings to human nutritional responses is vague without carefully controlled nutritional amino acid titration experiments using humans.

Most of the studies here were conducted using young (i.e., 8 weeks) male mice fed for a short period of time (i.e., 3 weeks). Thus, this may limit the applicability of the findings, especially since female mice can respond differently to dietary challenges[51–53] and AA requirements can be different during growth/maturation versus adulthood[50]. However, the majority of the effects of dietary AA restriction seen in young mice (i.e., feed efficiency, energy expenditure, food intake, glucose metabolism) were also reflected in older male and female mice, albeit with female mice responding to a lesser extent (Fig. 4). Of note however, were the effects of the different AA restriction diets on lean body mass and skeletal/heart muscle mass with total AA or EAA restriction lowering lean tissue mass, whereas the low Thr diet did not. This highlights that a low Thr diet can induce many of the positive effects of dietary AA restriction while avoiding some negative side effects such as lean tissue wasting. On this, similar to that previously shown for dietary protein/AA dilution[14,16,21,49,68], and SCAA or BCAA restriction[21,38,42–44], low Thr feeding also retarded development of obesity-induced metabolic dysfunction in a mouse model (Fig. 5). Dietary Thr restriction may thus be an attractive strategy to mimic the effects of DPD without unwanted caveats, particularly as dietary Thr is largely metabolised by the intestine[69] and thus many potential undesirable systemic side-effects may be avoided.

Several studies have demonstrated that systemic[14,16,24,25,32,40,44,68] or liver-restricted[14,26] Fgf21-loss retards the downstream metabolic effects of dietary protein/AA restriction. In particular, these effects are likely to be mediated by central nervous system FGF21 signalling[68], to stimulate nutrient uptake/oxidation in adipose tissue and heart[14,16]. Here we extend upon

these findings and demonstrate that adult, liver-specific FGF21 loss completely abrogates the systemic metabolic remodelling with dietary Thr restriction (Fig. 6). In addition, given that liver/hepatocyte-specific restoration of EAA biosynthesis could rescue effects of EAA restriction (Fig. 7), taken together this reinforces the prior evidence[14] that the liver is the chief anatomical site "sensing" dietary protein/AA restriction which is logical given its proximal anatomical position to nutrient absorption.

In conclusion, from studies of mice using a casein-based diets, the restriction of EAA, particularly Thr and Trp, are sufficient and necessary to confer the systemic metabolic effects of DPD.

## Methods

All unique materials are available from commercial vendors or from the authors pending material transfer agreements. The data that support the findings of this study are available from the corresponding author upon reasonable request.

**Diets**. Diets were purchased either from Specialty Feeds (Perth, Australia; Supplementary Table 1) or Research Diets (NJ, USA; Supplementary Table 2). Study A (Fig. 1 and Supplementary Fig. 1) was a comparison of diets SF17-180 (20% E from protein), SF17-175 (5% E from protein), and SF17-176 (5% E from protein and 15% E from amino acids). Study B (Fig. 2a, b) was a comparison of diets D12450B (20% E protein) and D10062201 (5% E protein). Study C (Fig. 2c–i and Supplementary Fig. 2) was a comparison of amino acid containing diets A14011601–A14011606. Study D (Fig. 3a–d and Supplementary Fig. 3A–E) was a comparison of diets A14011601, A14011605, and A16120501–A16120503. Study E (Supplementary Fig. 3F–I) was a comparison of diets A14011601, A14011605, and A17020901–A17020903. Study F (Fig. 3e–h and Supplementary Fig. 3J–N) was a comparison of diets A14011601, A14011605, and A170401301–A170401303. Study G (Supplementary Fig. 3O–R) was a comparison of diets SF14-162 and SF17-114. Study H (Fig. 3i–l and Supplementary Fig. 3S–W) was a comparison of diets SF17-177, SF18-109, and SF17-110. Study I (Fig. 4 and Supplementary Fig. 4) was a comparison of diets SF17-177, SF18-109, SF19-086, and SF17-179. Study J (Fig. 5e–g and Supplementary Fig. 5) and study K (Fig. 6 and Supplementary Fig. 6) was a comparison of diets SF17-177 and SF17-179. Study K (Fig. 7 and Supplementary Fig. 7) was a comparison of diets SF17-177 and SF18-066.

**Recombinant viruses**. A control (green fluorescent protein: GFP), Cre-recombinase, yeast (*Saccharomyces cerevisiae*) threonine 1 (Thr1), or threonine 4 (Thr4) cDNAs were cloned into AAV genome plasmids for subsequent production of self-complimentary adeno-associated viruses (AAV) using the three plasmid system. The genome plasmid encoding either hepatocyte-specific regulated (via LP1-promoter[70,71]) GFP, Thr1, or Thr4 was co-transfected with the adenoviral helper plasmid pDGΔVP and the capsid plasmid p5E18VD2/8-mut6 (aa 589–592: QNTA to GNRQ; AAV8 to AAV2)[72]. Purification and quantification of the final vector stocks was done by density gradient/filtration purification and quantitative real-time PCR, respectively[73].

**Mouse experiments**. Unless stated otherwise, male mice aged 7 weeks upon arrival, were acclimated to the local housing facility (12–12 h light–dark cycle, 22–24 °C) for 1 week prior to experimentation and were fed standard rodent chow (3437, Provimi Kliba, Germany or 8720610, Barastoc, Australia) unless otherwise indicated. Mice used for experiments were C57Bl/6NCrl mice (#027, Charles River Laboratories, Germany), C57Bl/6J (Monash University Animal Research Platform, Clayton, Australia or Animal Resource Centre, Australia). Slc6a19−/− and corresponding +/+ littermates[22] as well as Fgf21 fl/fl littermate mice[74] were also used, both on C57Bl/6J background. New Zealand Obese mice[75], a model of obesity-induced type 2 diabetes, were also used.

The dietary intervention was identical for nearly all experiments (Figs. 1–3 and 7) and is outlined in Supplementary Fig. 1A. In brief, following acclimation, mice were placed on diets for 3 weeks with body weight recorded each week and metabolic cage housing during week 2 with a 5-h fasting bleed (sometimes with a glucose tolerance test) conducted during week 3 (i.e., day 19–20). Mice were then humanely euthanised for tissue collection.

To examine the chronic effects of dietary AA restriction on fully developed mice, 6-month-old male and female C57Bl/6J mice were treated with diets SF17-177 (NAA), SF18-109 (LAA), SF19-086 (LEAA), and SF17-179 (LT) for a period of 8 weeks. Body weight and composition (ECHO-MRI™ 3in1, EchoMRI LLC, USA) was measured before and at the end of the treatment period. Metabolic cage housing was completed for 5 days, 1-week after diet initiation. A 5 h fasting bleed was conducted 7 weeks after diet initiation. At the end of 8 weeks, mice were humanely euthanised for tissue collection.

For studies on New Zealand Obese mice, mice were obtained at 7 weeks of age and acclimated to the housing facility. At 8 weeks, they were switched to either diet SF17-177 (NAA) or SF17-179 (LT) and studies for a further 4 weeks. Mice were weighed and bled for blood glucose measurement in the random fed state before,

and each week. At the end of 8 weeks, mice were humanely euthanised for tissue collection.

For liver/hepatocyte-specific Fgf21 silencing experiments (Fig. 6), we conducted experiments where following acclimation, 7-week-old Fgf21fl/fl mice were administered a total of $2.5 \times 10^{11}$ virus particles per mouse via the tail vein. For the negative control (NC): $2.5 \times 10^{11}$ GFP-AAV; and for the Cre-recombinase overexpression studies mice were administered $2.5 \times 10^{11}$ virus particles each of CRE-AAV. One week following this time, the dietary intervention was initiated and continued for 8 weeks with metabolic cage housing during week 2 and a fasting bleed during week 7. At the end of 8 weeks, mice were humanely euthanised for tissue collection.

For liver/hepatocyte-specific yTHR1 and yTHR4 expression experiments (Fig. 7), we conducted experiments where following acclimation, mice were administered a total of $5 \times 10^{11}$ virus particles per mouse via the tail vein. For the negative control (NC): $5 \times 10^{11}$ GFP-AAV; and for the *THR1/4* overexpression studies mice were administered $2.5 \times 10^{11}$ virus particles each of yTHR1-AAV and yTHR4-AAV. One week following this time, the dietary intervention was initiated. In these studies, the low threonine diets contained homoserine, the substrate of THR1. Importantly, pilot studies showed no differences in the response to a low threonine diet with or without supplemented homoserine.

For the experiment involving selective restriction of Met, Thr, and Trp, mice were placed on one of two experimental diets at 12 weeks of age for 6 weeks. Food intake was measured weekly and body weights were measured every 2 weeks. Animals were then humanely euthanised for tissue collection.

Fasting–refeeding experiments were conducted to nutritionally synchronise mice for metabolomics measurement. As such, cohorts of mice was adapted to a control or low protein diet for 1 week using diets described[14]. After this, they were fasted overnight, and then refed the same diet for 4–5 h following which they were anesthetised for hepatic portal vein bleeding (cohort 1) or killed by cervical dislocation for rapid freezing of the liver *in situ* using a freeze clamp precooled in $LN_2$ (cohort 2).

We conducted hepatic portal vein bleeding from Slc6a19−/− and corresponding +/+ littermate mice with blood serum amino acid profiling, including sample preparation and derivatization, by LC–MS/MS using the EZ:faast kit (Phenomenex)[14]. In order to test for the requirement of systemic Thr and Trp lowering on serum FGF21 in the background of Slc6a19 loss-of-function, we fasted Slc6a19−/− and corresponding +/+ littermates overnight and then refed mice a 20% EP diet for 2 h, after which we withdrew the food. Upon food withdrawal, mice were intraperitoneally injected a 0.9% saline solution (Vehicle), or a mixture of L-threonine and L-tryptophan (6 mg each, 12 mg; ~0.5 mg/g body mass), or a mixture of L-phenylalanine and L-histidine (6 mg each, 12 mg; ~0.5 mg/g body mass) and tail vein blood was then collected 4–5 h later.

Animal experiments were conducted according to regional, national, and continental ethical guidelines and protocols were approved by local regulatory authorities (Regierungspräsidium Karlsruhe, Germany; Monash University Animal Ethics Committee, Australia; and University of Sydney Animal Ethics Committee, Australia) and conformed to ARRIVE guidelines.

**Biometric and metabolic phenotyping**. The dietary intervention study is outlined in Supplementary Fig. 1A. Mice were housed in groups of 2–3 unless placed within the metabolic cages (see below). Body weight was measured before and after the dietary intervention period and body mass difference was calculated. Indirect calorimetry and food intake was recorded by individual housing in metabolic phenotyping system cages (TSE Phenomaster System (TSE Systems, Germany) or Promethion-M High Definition Multiplexed Respirometry System (Sable Systems International, USA)). Energy expenditure (W) was estimated using the Weir equation[76] from $VO_2$ and $VCO_2$ measurements (EE (W or J/s) = ((1.44 × (3.94 × $VO_2$ (mL/h) + 1.11 × $VCO_2$ (mL/h))/1000 × 4.196) × 0.28). Feed efficiency was calculated from the quotient of the change in body mass (mg) and cumulative food energy intake (kJ) during the metabolic cage housing. To assess whole body glucose homoeostasis, a 5–6 h fasting blood sample (fasting initiated at ZT3, sampling at ZT8-9) was collected from the tail vein from which blood glucose (AccuCheck Aviva) and plasma insulin (80-INSMS, Alpco, USA) was measured. In some studies, an intraperitoneal glucose tolerance test was conducted immediately following the fasting blood sample by injection of a fixed dose of 50 mg of D-glucose in 0.9% saline[77] with blood samples drawn for glucose and insulin measurement at selected times after. In addition, HOMA-IR ((glucose (mM) × insulin (pM))/3857)[78] and a fasting insulin sensitivity index (ISIf) was calculated, ISI(f): 1000/(glucose (mM) × insulin (pM)), and are good surrogate indices of whole-body glucose homoeostasis[14,79]. The methods for assessment of glucose homoeostasis in mice were conducted in accordance with published guidelines[77,80]. Serum FGF21 (MF2100, R&D Systems, USA), urea (Z5030016, Biochain, USA), and triglycerides (TR0100, Sigma-Aldrich, USA) were also measured from the blood serum samples upon sacrifice by cervical dislocation between ZT3-5.

**Metabolomics**. Livers were cryogenically pulverised (cryopulverization) using a 12-well biopulverizer (BioSpec Products, OK, USA, Part number 59012MS) according to the manufacturer's instructions. The frozen tissue powder was then weighed and extracted in 20 μL of extraction solvent (0 °C) per mg of tissue. The mixture was then briefly vortexed before sonication in an ice-water bath for 10 min

followed by centrifugation (20,000 rcf, 4 °C, 10 min). The supernatant was then transferred to a mass spectrometry vial for LC–MS analysis. The extraction solvent consisted of 2:6:1 CHCl₃:MeOH:H₂O v/v/v with 2 μM CHAPS, CAPS, PIPES, and TRIS as internal standards. Additionally where quantitative amino acid analysis was performed, a mixture of stable isotope labelled amino acids were added at a concentration of 500 pmol of each amino acid per mg liver (Cambridge Isotope Laboratories PN MSK-A2-1.2).

Plasma (25 μL) was extracted by addition of 200 μL of 1:1 acetonitrile:MeOH v/ v with 1 μM CHAPS, CAPS, PIPES, and TRIS as internal standards at 0 °C. Samples were then mixed on a vortex mixer for 30 min at 4 °C after which they were centrifuged (20,000 rcf, 4 °C, 15 min) and the supernatant then transferred to a mass spectrometry vial for LC–MS analysis.

For LC–MS analysis, samples were analysed by hydrophilic interaction liquid chromatography coupled to high-resolution mass spectrometry (LC–MS)[81]. In brief, the chromatography utilised a ZIC-pHILIC column (column temperature 25 °C) with a gradient elution of 20 mM ammonium carbonate (A) and acetonitrile (B) (linear-gradient time—%B as follows: 0 min—80%, 15 min—50%, 18 min—5%, 21 min—5%, 24 min—80%, 32 min—80%) on a Dionex RSLC3000 UHPLC (Thermo). The flow rate was maintained at 300 μL/min. Samples were kept at 4 °C in the autosampler and 10 μL injected for analysis. The mass spectrometry was performed at 35,000 resolution (accuracy calibrated to <1 ppm) on a Q-Exactive Orbitrap MS (Thermo) operating in rapid switching positive (4 kV) and negative (−3.5 kV) mode electrospray ionisation (capillary temperature 300 °C; sheath gas 50; Aux gas 20; sweep gas 2; probe temp 120 °C). All samples were analysed in randomised order and with pooled quality control samples analysed regularly throughout the batch to confirm reproducibility. ~300 Metabolite standards, including all reported amino acids, were analysed immediately preceding the batch to determine accurate retention times to confirm metabolite identification.

For data analysis, untargeted metabolomics data were analysed using the IDEOM (version 20) workflow with default parameters[82]. In brief, this involved peak picking with XCMS[83], peak alignment and filtering with mzMatch[84] and further filtering, metabolite identification, and comparative analysis with IDEOM. Amino acid concentration was determined by the integration of the extracted ion chromatograms of the amino acids and their corresponding stable isotope labelled isotopologues in MZmine 2.32[85,86]. All peaks were inspected and where necessary the integration parameters altered to insure accurate integrations. Nearly all the amino acids were detected in both polarities, however we elected to use the negative mode data for all amino acids except Ala, Arg, Gly, and His where we used positive mode data. The amino acid concentration was then calculated by comparison of the peak areas of each amino acid against its corresponding heavy labelled isotopologue.

**Proteomics.** The mouse liver tissue was homogenised in liquid nitrogen using a BioPulverizer and directly solubilised in sodium dodecyl sulfate (SDS) lysis buffer. The protein concentration was determined using a BCA kit (Thermo Scientific) and equal amounts of protein were processed for both pooled control (GFP) and THR1/4-overexpressed mouse liver sample. SDS was removed by chloroform/ methanol precipitation and the proteins were proteolytically digested with trypsin (Promega) and purified using OMIX C18 Mini-Bed tips (Agilent Technologies) prior to LC–MS/MS analysis. Using a Dionex UltiMate 3000 RSLCnano system equipped with a Dionex UltiMate 3000 RS autosampler, an Acclaim PepMap RSLC analytical column (75 μm × 50 cm, nanoViper, C18, 2 μm, 100 Å; Thermo Scientific) and an Acclaim PepMap 100 trap column (100 μm × 2 cm, nanoViper, C18, 5 μm, 100 Å; Thermo Scientific), the tryptic peptides were separated by increasing concentrations of 80% ACN/0.1% formic acid at a flow of 250 nL/min for 158 min and analysed with a QExactive Plus mass spectrometer (Thermo Scientific) using in-house optimised parameters to maximise the number of peptide identifications. To obtain peptide sequence information, the raw files were searched with Byonic v3.0.0 (ProteinMetrics) against a mouse UniProt/SwissProt database that was appended with the yeast THR1/THR4 protein sequences. Only proteins falling within a false discovery rate of 1% based on a decoy database were considered for further analysis.

**RNA extraction and analysis.** RNA was extracted from tissues using QIAzol and cDNA was synthesised using the Quantitect Reverse Transcription Kit (Qiagen). qPCR was conducted using Quantitect Sybr Green qPCR (Qiagen) with the following primers: PrimePCR™ SYBR® Green Assays to yeast *Thr1* (qSceCED0001111, Bio-Rad) and *Thr4* (qSceCED0005815, Bio-Rad) as well as mouse *Tbp* (QT00198443, Qiagen) as a housekeeping gene.

**Statistical analyses.** Mice were assigned to groups based upon initial body mass for counterbalancing. Pre-established criteria for exclusion of mice from study groups were obvious infections/wounds which would impact on feeding behaviour as well as metabolic profile. Where possible, analysis of data collection was blinded.

Statistical analyses were performed using *t*-tests (two-sided), or 2-way analysis of variance (ANOVA) with or without repeated measures, where appropriate, with Holm–Sidak-adjusted post-tests. All analyses were carried out with GraphPad Prism v.7.01 (GraphPad Software, Inc.) or SigmaPlot 14 (Systat Software, Inc.)

software. Statistical details can be found within the figure legends. Differences between groups were considered significant when $P < 0.05$.

**Reporting summary.** Further information on research design is available in the Nature Research Reporting Summary linked to this article.

## Data availability

The source data underlying Figs. 1a–i, 2a, b, d–j, 3a–l, 4a–l, 5a–e, 6a–d, and 7a–d and Supplementary Figs. 1B–F, 2A–L, 3A–W, 4A–L, 5A, B, 6A–H, and 7A–F are provided as a Source Data file. The annotated metabolomics and proteomics data can be found within Supplementary Datasets 1 and 2, respectively. The metabolomics data is available at the NIH Common Fund's National Metabolomics Data Repository (NMDR) website, the Metabolomics Workbench, https://www.metabolomicsworkbench.org where it has been assigned Project ID PR000917. The data can be accessed directly via its Project DOI: 10.21228/M8568J. The mass spectrometry proteomics data have been deposited to the ProteomeXchange Consortium via the PRIDE[87] partner repository with the dataset identifier PXD018205.

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

## Acknowledgements

The authors wish to thank former lab members Thomas Gantert and Adriano Maida (German Cancer Research Center, Heidelberg, Germany) who conducted the pilot experiments for this study series. The experimental and technical assistance of Robert Lee (Monash Metabolic Phenotyping Facility, Australia), Anja Pfenninger (Sanofi-Aventis, Germany), Annika Zota, Max Schuster, and Sarah Lerch (German Cancer Research Center, Germany) is gratefully acknowledged. These studies were supported by Monash Biomedicine Discovery Institute laboratory start-up funds to A.J.R. as well as a Monash Joint Science-Medicine Interdisciplinary Research Seed Funding to A.J.R. and M.D.W.P.

## Author contributions

Study conception: A.J.R. Study design: A.J.R. Reagent design, construction, and provision: B.C.F., A.J., B.M., S.B., S.A., O.J.M., A.J.R. Performed experiments: Y.W.Y., P.M.R., A.Y.C., S.M.S.-B., B.K., M.H., A.J.R. Analysed samples and data: Y.W.Y., S.M.S.-B., C.K.B., D.J.C., C.H., R.B.S., D.S., B.K., M.D.W.P., M.H., S.J.S., A.J.R. Wrote manuscript: A.J.R. Drafted manuscript: Y.W.Y., P.M.R., S.M.S.-B., C.K.B., C.H., M.D.W.P., S.J.S., S.B., O.J.M., A.J.R.

## Competing interests

D.S. is an employee of Sanofi-Aventis Deutschland GmbH, a pharmaceutical company. All other authors declare no competing interests.
