## [Peer Review File · Nature Communications]

Reviewers' Comments:

Reviewer #1:

Remarks to the Author:

No further comments

Reviewer #5:

Remarks to the Author:

As originally stated, this manuscript by Yap and colleagues describes a thorough and interesting set of experiments that seek to understand the specific contributions of essential and non-essential amino acids to dietary protein dilution (DPD). This is an area of emerging importance as protein restriction is emerging as a promising strategy for the treatment of diseases ranging from diabetes to Alzheimer's diseases. Here, the authors specifically identify the essential amino acids threonine and tryptophan (and not other groups of essential amino acids or essential lysine), and demonstrate that specific restriction of these amino acids are necessary and sufficient to recapitulate the systemic metabolic response to DPD in mice. This research combines a novel study design to elucidate the complicated relationship between macronutrient composition and specific amino acid restriction, and uses genetic tools to demonstrate the critical role of threonine in the metabolic response to DPD.

Overall, the revised manuscript still represents a timely and important contribution to the fields of nutrition and metabolism; and the reviewers have made major progress in revising and addressing the points of this reviewer as well as my colleagues. However, as discussed below, there remain a few unaddressed points that should be addressed prior to finalizing the manuscript.

1. As noted originally, "I think the authors need to reword the abstract (in particular) to highlight the context-dependence of these results, based on their findings that these amino acids, particularly Thr, are likely the most limiting AA in a casein based diet. This would appropriately limit the results to the present context and link to protein source – as a casein based diet does not represent the context of all of the experimental literature, or necessarily the human diet"

While the authors have acknowledged that this is an important issue and state that they "agree and have softened our stance that it is only these essential amino acids that would be most limiting under all circumstances of dietary protein restriction. We have include additional references and some more discussion around this point."... the reviewer believes any reader of the manuscript or abstract would not realize that anything has been softened.

As noted originally - obviously Thr restriction is not ALWAYS the key AA as ample literature has identified methionine and other AAs as necessary and sufficient to convey the effects of DPD in other contexts (this other research includes some previous work of the corresponding author). Thus, the abstract and discussion should be revised more substantially to clarify the novelty and limitations of the research.

2. Regarding metabolic indispensibility, this reviewer previously noted that "while technically correct it is likely irrelevant in the context of the rodent diets fed here (or in the human diet); there are no metabolically significant quantities of these alpha keto acids in the diet, and thus no quantities of these alpha keto acids in the mouse that did not derive originally from EAAs; and

thus all of the EAAs here are equally metabolically indispensable. Thus, the conclusion that Thr, Trp and Lys are critically important in this context due to their metabolically indispensable nature should be reconsidered."

The authors state: "We are not entirely sure that such precursors are at such levels to be metabolically inert. For example, the liver hepatocytes do not express a branched chain amino acid transaminase but substantive metabolism of these amino acids still occurs in the liver (PMID: 30449684)."

The authors should read 30449684 closely; per citation to a recent proteomic atlas of the liver, the NON-hepatocyte cells of the liver do express BCAT, thus reconciling Dr. Hutson's research with that of Dr. Arany and other much older radiotracing experiments. There remains no evidence the reviewer is aware of that dietary α -ketoacids are a relevant source of BCAAs and this section regarding indispensability should therefore be reworded to clarify that this is a limitation.

Reviewer #6:

Remarks to the Author:

The instance on the use of the term dietary protein dilution is odd...this is not a well accepted term.

In many places the authors try to make this work relevant for humans-and even measure human samples- hence the authors should consider the use of the word indispensable instead of EAA....most people working in the area of human nutrition and diet use these terms.

What I miss completely is a mention/acknowledgment of nitrogen balance- this is fundamental to protein intake. There are well established relationships between nitrogen balance , energy intake and protein intake that are not mentioned here- this is fundamental to any work in humans.

Well established that the level of energy intake determines the degree of change in nitrogen balance that occurs in response to a change in nitrogen intake. Conversely, the level of nitrogen intake determines the quantitative effect of energy intake on nitrogen balance.

To make this work truly innovative and informative it is essential to do tracer studies.

Very well conducted experiments- but the relevance to human is minimal without the correct study of nitrogen balance

Response to reviewers

REVIEWERS' COMMENTS:

Reviewer #1 (Remarks to the Author):

No further comments

We thank this reviewer for their time and efforts to carefully review our work.

Reviewer #5 (Remarks to the Author):

As originally stated, this manuscript by Yap and colleagues describes a thorough and interesting set of experiments that seek to understand the specific contributions of essential and non-essential amino acids to dietary protein dilution (DPD). This is an area of emerging importance as protein restriction is emerging as a promising strategy for the treatment of diseases ranging from diabetes to Alzheimer's diseases. Here, the authors specifically identify the essential amino acids threonine and tryptophan (and not other groups of essential amino acids or essential lysine), and demonstrate that specific restriction of these amino acids are necessary and sufficient to recapitulate the systemic metabolic response to DPD in mice. This research combines a novel study design to elucidate the complicated relationship between macronutrient composition and specific amino acid restriction, and uses genetic tools to demonstrate the critical role of threonine in the metabolic response to DPD.

Overall, the revised manuscript still represents a timely and important contribution to the fields of nutrition and metabolism; and the reviewers have made major progress in revising and addressing the points of this reviewer as well as my colleagues. However, as discussed below, there remain a few unaddressed points that should be addressed prior to finalizing the manuscript.

We thank this reviewer for their appreciation of our studies and their time and efforts to carefully review our work.

1. As noted originally, "I think the authors need to reword the abstract (in particular) to highlight the context-dependence of these results, based on their findings that these amino acids, particularly Thr, are likely the most limiting AA in a casein based diet. This would appropriately limit the results to the present context and link to protein source – as a casein based diet does not represent the context of all of the experimental literature, or necessarily the human diet"

While the authors have acknowledged that this is an important issue and state that they "agree and have softened our stance that it is only these essential amino acids that would be most limiting under all circumstances of dietary protein restriction. We have include additional references and some more discussion around this point."... the reviewer believes any reader of the manuscript or abstract would not realize that anything has been softened.

As noted originally - obviously Thr restriction is not ALWAYS the key AA as ample literature has identified methionine and other AAs as necessary and sufficient to convey the effects of DPD in other contexts (this other research includes some previous work of the corresponding author). Thus,

the abstract and discussion should be revised more substantially to clarify the novelty and limitations of the research.

These are fair points. We have reworded the abstract to read: ..."by mimicking amino acid (AA) supply from a casein-based diet...". While we would like to have edited further, we are limited to 150 words, and unfortunately, we do not have any extra space. By placing this statement at the start of the abstract, we feel that the reader will then be able to place the following statements in this context.

Within the discussion, we have clearly noted that AAs other than K, T and W can be provided within a diet at sufficiently low levels to induce FGF21 and associated metabolic remodelling and cited evidence for this. We thought it an interesting discussion point that not all EAAs are equal in this context and that some may even be more limiting than others according to a relationship between dietary AA supply and a certain somatic variable such as the rate of AA metabolism. To our knowledge, this remains to be formally tested and we have noted this in the discussion (see below).

2. Regarding metabolic indispensibility, this reviewer previously noted that "while technically correct it is likely irrelevant in the context of the rodent diets fed here (or in the human diet); there are no metabolically significant quantities of these alpha keto acids in the diet, and thus no quantities of these alpha keto acids in the mouse that did not derive originally from EAAs; and thus all of the EAAs here are equally metabolically indispensable. Thus, the conclusion that Thr, Trp and Lys are critically important in this context due to their metabolically indispensable nature should be reconsidered." The authors state: "We are not entirely sure that such precursors are at such levels to be metabolically inert. For example, the liver hepatocytes do not express a branched chain amino acid transaminase but substantive metabolism of these amino acids still occurs in the liver (PMID: 30449684)."

The authors should read 30449684 closely; per citation to a recent proteomic atlas of the liver, the NON-hepatocyte cells of the liver do express BCAT, thus reconciling Dr. Hutson's research with that of Dr. Arany and other much older radiotracing experiments. There remains no evidence the reviewer is aware of that dietary a-ketoacids are a relevant source of BCAAs and this section regarding indispensibility should therefore be reworded to clarify that this is a limitation.

This is an important point. We completely agree that it is highly unlikely that there are metabolically significant quantities of these alpha ketoacids in a natural diet. However, this was not our point. Our point is that metabolite sharing and cycling between tissues can potentially overcome a dietary shortfall of a certain AA. We have read the work of both Dr. Arany and Dr. Hutson carefully, and yes, substantive metabolism of BCAA can occur in the liver in vivo even though it expresses very little BCAT. Yes, these non-parenchymal cells do express BCAT (with very little to nil expression in hepatocytes), but this pales in comparison to brain, skeletal muscle and some other tissues, and very little BCAAs are catabolised/cleared on first pass through the liver which validates this point. Thus, the breakdown products of BCAAs from other cell types (including liver non-parenchymal cells) probably feed the liver hepatocytes and this must occur very rapidly in vivo as demonstrated by the recent studies from Dr. Arany and Dr. Rabinowitz as well as former studies using BCKA tracers. Thus, we contend that as there can be metabolic sparing from AAs which share a similar metabolic reactions (e.g. Met/Cys and BCAAs) and we have included a more detailed discussion point around this. Furthermore, we have clarified our stance further within the discussion to state: *"Thus we propose that it is the strictly metabolically EAA, namely Lys, Thr and Trp, which will have the potential to be most limiting in a particular diet. Nonetheless, the*

differences in the sensitivity of AA restriction requires careful studies of nutritional EAA titration, especially as or studies were fully based on AA supply from a single nutritional source (i.e. the milk protein casein). Thus, this is a clear direction for future studies."

Reviewer #6 (Remarks to the Author):

We thank this reviewer for their time and efforts to carefully review our work.

The instance on the use of the term dietary protein dilution is odd...this is not a well accepted term. In many places the authors try to make this work relevant for humans-and even measure human samples- hence the authors should consider the use of the word indispensable instead of EAA....most people working in the area of human nutrition and diet use these terms.

In an isocaloric diet when you change one macronutrient source concentration, another macronutrient must be also altered in turn thus altering the ratio (dilution or enrichment). Thus, we use the term dilution as it is simply more technically correct. However, as stated previously, at least from what we see from our data, it does appear that the restriction of certain AAs is the factor that matters.

The use of the term "essential" versus "indispensable" is a stylistic point and thus we do not feel that this needs to be changed.

What I miss completely is a mention/acknowledgment of nitrogen balance- this is fundamental to protein intake. There are well established relationships between nitrogen balance , energy intake and protein intake that are not mentioned here- this is fundamental to any work in humans. Well established that the level of energy intake determines the degree of change in nitrogen balance that occurs in response to a change in nitrogen intake. Conversely, the level of nitrogen intake determines the quantitative effect of energy intake on nitrogen balance. To make this work truly innovative and informative it is essential to do tracer studies. Very well conducted experiments- but the relevance to human is minimal without the correct study of nitrogen balance

While we understand that nitrogen balance is an important aspect of nutrition and protein-nutrition, we are not convinced that it matters in the context of our observed responses. In particular, in Figure 2 we have now included data on urinary urea output, a proxy for total protein oxidation, and we could separate the effects of total dietary protein oxidation and FGF21 response and related systemic metabolic remodelling, in response to altered EAA/NEAA supply. Although it may not apply to our studies here of ad libitum fed mice, the relationships between total energy and nitrogen intake on responses are however indeed interesting and we have included this discussion point "our studies here (Fig. 1-7), as well as previous studies of amino acid (AA) restriction 23,37, have conclusively shown that a restriction of dietary protein/AA per se can induce the systemic metabolic response to dietary protein dilution. This occurs even when total dietary AA enrichment is held constant (Fig. 1-7), and total protein oxidation is not affected (Fig. 2D). Nevertheless, there are important relationships between total energy and N intake 6,57, and deciphering the roles of certain AAs and FGF21 in this context will be interesting." We would be interested to hear this reviewers further thoughts on this issue and the proper tracer studies

required to investigate this. Of further note, we have removed the human data from the manuscript, and added the statement *“as our studies were conducted in mice the relevance of our findings to human nutritional responses is vague without carefully controlled nutritional amino acid titration experiments using humans.”*